# IC-Based Neuro-Stimulation Environment for Arbitrary Waveform Generation

**Florian Kolbl** [1,*] , **Yannick Bornat** [2] , **Jonathan Castelli** [2] , **Louis Regnacq** [1,2] , **Gilles N'Kaoua** [2] , **Sylvie Renaud** [2] and **Noëlle Lewis** [2]

[1] ETIS CNRS UMR 8051, CY Cergy Paris University, ENSEA, 95000 Cergy, France; louis.regnacq@ensea.fr
[2] IMS, Université de Bordeaux, CNRS UMR 5218, Bordeaux INP, 33400 Talence, France;
yannick.bornat@ims-bordeaux.fr (Y.B.); castelli.jonathan@gmail.com (J.C.);
gilles.nkaoua@u-bordeaux.fr (G.N.); sylvie.renaud@ims-bordeaux.fr (S.R.);
noelle.lewis@ims-bordeaux.fr (N.L.)
* Correspondence: florian.kolbl@ensea.fr

**Abstract:** Electrical stimulation of the nervous system is commonly based on biphasic stimulation waveforms, which limits its relevance for some applications, such as selective stimulation. We propose in this paper a stimulator capable of delivering arbitrary waveforms to electrodes, and suitable for non-conventional stimulation strategies. Such a system enables in vivo stimulation protocols with optimized efficacy or energy efficiency. The designed system comprises a High Voltage CMOS ASIC generating a configurable stimulating current, driven by a digital circuitry implemented on a FPGA. After fabrication, the ASIC and system were characterized and tested; they successfully generated programmable waveforms with a frequential content up to 1.2 MHz and a voltage compliance between $[-17.9; +18.3]$ V. The system is not optimum when compared to single application stimulators, but no embedded stimulator in the literature offers an equivalent bandwidth which allows the wide range of stimulation paradigms, including high-frequency blocking stimulation. We consider that this stimulator will help test unconventional stimulation waveforms and can be used to generate proof-of-concept data before designing implantable and application-dedicated implantable stimulators.

**Keywords:** biomedical electronics; electrical stimulation; neurostimulation; biomedical engineering

## 1. Introduction

A growing number of diseases and disabilities are treated using active implantable medical devices. Among them, neuro-prostheses are based on electrical stimulation of the Peripheral or Central Nervous System (PNS or CNS) to enhance cognitive, motor, or sensory abilities. Figure 1a illustrates the range of therapeutic applications, within three main categories.

A first category regroups sensory feedback applications, such as cochlear implant [1]—probably the most commonly implanted PNS stimulation hardware and retinal [2] or vestibular [3] prostheses. More recently, somatosensory feedback restoration for patients suffering from limb damage has been investigated through electrical stimulation [4]. Motor control prostheses form a second major group of applications for electrical stimulation. Among them Functional Electrical Stimulation (FES) is one kind of stimulation aiming at movement rehabilitation after lesions such as spinal cord injuries. Movement restoration can be achieved in some cases, as summarized in [5]. Other applications aim at restoring some involuntary and visceral motor functions. For example, sacral stimulation can be used in case of faecal [6] or urinary [7] incontinence after spinal cord injuries. Another example is the control of the respiratory system after spinal cord injury using efferent PNS stimulation, as explained in [8]. A third category deals with Central Nervous System (CNS) applications. It can be divided into two main sub-categories: stimulation of the brain

and of the spinal cord. Stimulation of the spinal cord is already successfully used for the treatment of chronic pain, as explained in [9]. Recent advances also suggest that spinal cord stimulation could be an interesting approach for movement rehabilitation, as explained in [10]. The stimulation of brain structures also leads to different applications. Deep Brain Stimulation (DBS) is now a well-recognized technique for neurological pathologies such as Parkinson's disease [11], chorea [12] or depression [13]. With DBS, the stimulation targets are deep nuclei structures of the brain, although the underlying mechanisms have not been fully identified. Other structures can be targeted, for example the auditory cortex for suppression of tinnitus, as explained in [14]. Note that a last class of applications can be related to indirect CNS stimulation, like vagus nerve stimulation used to treat epilepsy [15] for instance.

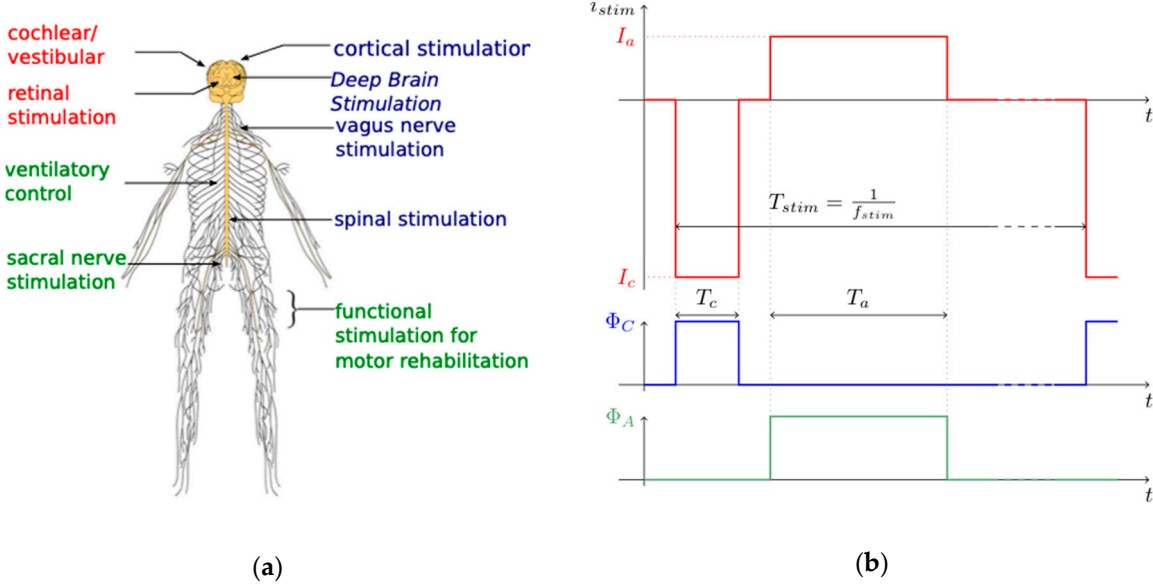

(**a**)  (**b**)

**Figure 1.** (**a**) Illustration of some therapeutic applications for electrical stimulation along the nervous system: in red, sensory feedback applications; in green, motor control applications; and in blue, those for the central nervous system. (**b**) Typical biphasic current stimulation waveform with active balancing. The cathodic current *Ic* is negative and applied during a time *Tc* required to trigger excitable cells, and the injected charge is then balanced using a positive anodic pulse of duration *Ta*. Both pulses are generated with current sources activated by $\Phi_C$ and $\Phi_A$ switching commands.

Despite these extremely diverse physio-pathological contexts, the design of stimulation devices is usually driven by a common denominator: the stimulation biphasic waveform, as depicted in Figure 1b. All stimulators induce neural reactions with a similar technique, and targeted cells are excited or inhibited using specific electrodes. A typical biphasic current stimulation waveform is composed of two constant current pulses: first a cathodic, negative, current (*Ic*) pulse is delivered, lasting *Tc* to induce firing or inhibition of targeted cells; adequate *Ic* and *Tc* values are chosen by electro-physiologists or clinicians and depend on the electrode and the tissue properties. Then, an anodic, positive, current (*Ia*) pulse lasting *Ta* is provided, to avoid electrochemical reactions due to accumulation of injected charges that can damage tissue. Such a strategy is called active charge balancing [16]. Another alternative for charge balancing is a passive discharge of the electrode after the cathodic phase, by discharging the capacitance of the electrode. From a large perspective, stimulator devices are controllable current sources associated with switches activated for cathodic ($\Phi_C$) or anodic ($\Phi_A$) commands. Stimulation is applied at a given frequency using this pattern ($f_{stim}$) or using bursts of biphasic pulses.

Regardless of the clinical application, this stimulation paradigm has, however, been challenged on several of its features. We address in this paper non-conventional stimulation with a dedicated system enabling complex waveform generation. The entire setup has

been designed to perform stimulation on a large number of channels and enable experimentations in various applicative and clinical contexts. Our system generates various non-conventional waveforms reported in the literature. Their performance are discussed regarding the following features:

- Energy efficient waveforms: Biphasic current stimulation is performed on electrodes that are mainly capacitive. The associated voltage can reach tens of volts and the resulting amount of energy for stimulation can be high compared to the overall consumption of implants. To decrease the battery volume or increase its lifetime, modifications of the waveshape have been proposed. In [17], the authors used a linear model of the electrode to lower the stimulation voltage by subdividing the cathodic pulse and calculating successive values of stimulation current. In [18,19], computational models of the electrode impedance are used with optimization algorithms to produce biphasic exponential decaying pulses. Complex pulses including triangle, linear decay, gaussians or sinusoidal pulses are investigated in [20,21].
- Increased selectivity waveforms: The recruitment of identified cells or the activation of small groups of fiber has also been investigated. Modulating the shape of the cathodic pulse by taking into account specific properties of targeted cells, enables to discriminate them [22]. Techniques such as slow rising pulses, pre-pulse for cell depolarization or anodal block have also been used [23].
- High-frequency blocking stimulation (HFBS): Stimulation with frequencies above 1 kHz has been reported to block axonal activities [24]. This technique is used in various contexts [25,26] to block pain signals or unwanted direction propagation of neural signals with conventional stimulation. The waveform consists of a square or sinusoidal waveform with a frequency starting from 5 kHz to a maximum of 70 kHz [27] and limited by available stimulator performances [28], discriminating capabilities between groups of fiber occurs near this frequential limit [29].
- Biomimetic/controllable burst stimulation: The rehabilitation of complex activities requires accurate recruitment of large groups of cells. This is usually addressed by modulating the envelope of bursts of biphasic stimuli. This method was for example recently used to improve the naturalness of tactile sensory rehabilitation [30]. In the case of closed loop controlled therapeutic strategy such as respiratory control, this envelope can also be adapted in real-time to control physiological reaction [31].

All these waveforms, as opposed to biphasic waveforms, present a well-defined spectrum which requires a stimulator based on rapidly tunable current sources.

Conventional stimulators are designed to fulfil requirements imposed by the biphasic waveform. Circuits consist in one or two controllable currents sources associated with switches to impose accurate timings for anodic and cathodic pulses. Designers mainly face two challenges. On the one hand, high impedance electrodes with capacitive behavior require energy-efficient technologies enduring voltages up to tens of Volts. On the other hand, tissue safety imposes a well-controlled timings and relative precision of current sources when more than one is used: stimulators can be evaluated by their remaining charge or equivalent DC current. A literature review of stimulators has been set up in [32] and updated for this contribution. Figure 2a shows the evolution of technologies used for integrated stimulation circuits. Interestingly, for a decade, 0.35 μm and 18 μm technologies were mostly used, as their High Voltage processes reach the requirements imposed by electrode impedance. This review also discusses the ranking of stimulators for conventional stimulation by proposing a dedicated Figure of Merit (FOM, the lowest the best), integrating energy efficiency ($E_{ECS}$, the lowest the best) and tissue safety (SENOB, the lowest the best). The up-to-date version of the FOM ranking proposed in [32] is shown in Figure 2b. If technological improvements still occur in circuits for conventional stimulation, design challenges are clearly identified.

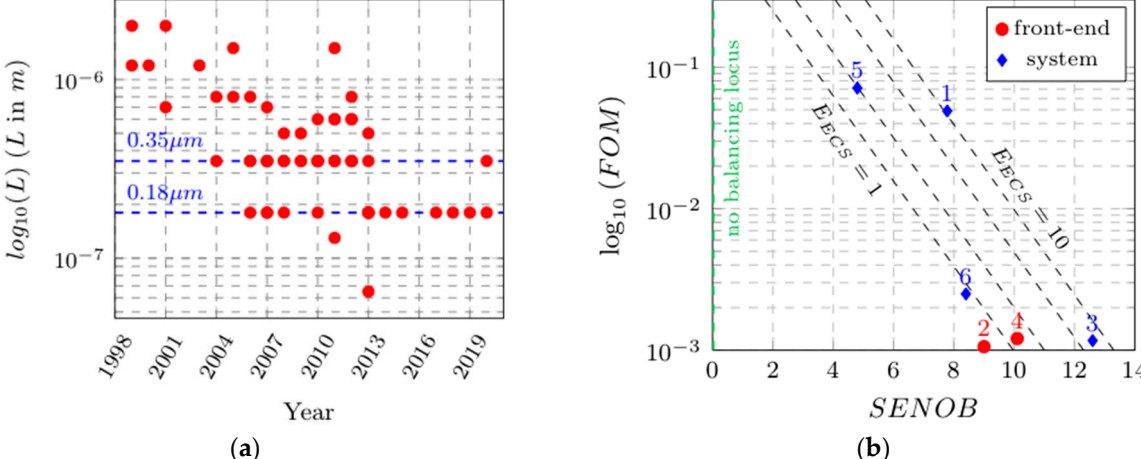

**Figure 2.** (**a**) Technological evolution (minimal technologic gate length) of stimulator circuits between 1998 and 2019. (**b**) FOM evaluation versus tissue safety criterion [32] of current trends in stimulation. (♯1: [33], ♯2: [34], ♯3: [35], ♯4: [36], ♯5: [37], ♯6: [38]).

Some neurostimulators have recently been designed for non-conventional waveforms. A large majority aim to improve energy efficiency, by replacing standard topology with switched mode DC–DC converter architecture. In [38], the authors propose an inductor-based buck-boost DC–DC converter which does not require an external output capacitor. This is made possible by replacing the biphasic waveform with high-frequency pulses. Similarly, [39] proposes an 'adiabatic' neurostimulator based on a dynamic power supply, which allows to remove the output H-bridge. Following a study aiming at finding the optimal energy efficient waveform [40], a fully implantable neurostimulator was designed in [41], based on the discovered waveform. In [42], the use of biphasic burst instead of conventional stimulation allows to integrate the capacitor needed for the power isolation inside the chip, reducing both footprint and cost of the system. More recently, with the growing interest in high frequency stimulation, a few stimulation systems have been designed specifically for this aim. In [43], the authors present a system based on commercially available components for high frequency blocking on four channels simultaneously, with a current stimulation up to 10 mA, and a frequency up to a few tens of kHz. Similarly, ref. [44] proposes a neurostimulator for high frequency blocking as well, based on an efficient switched mode DC–DC converter and equipped with an offset cancellation feedback circuitry. In [45], the authors present a structure capable of arbitrary waveform generation; however, dynamic performances are only discussed on low resistive loads and waveforms are externally generated.

In this contribution, we propose a stimulator bringing together current sources in an Application Specific Integrated Circuit (ASIC) chip, and a dedicated digital architecture enabling complex waveform generation at high frequencies. This stimulation system is intended to realize novel stimulation protocols with enhanced selectivity and efficacy (in terms of specific sub-populations stimulation) and optimized energy efficiency. We discuss and evaluate the performances of the proposed system for many waveforms and electrode types, i.e., for a wide range of applications.

## 2. Materials and Methods

### 2.1. Design Methodology: System Overview

The 8-channel stimulation system, named TWIST (arbiTrary Waveform STimulation system) is composed of analog current sources (on ASIC), a reconfigurable digital component (FPGA), power supplies and discrete components (DAC and voltage-to-current converters) to provide the low-voltage input currents required by the ASIC (Figure 3). This system can be controlled by the user via a software interface, or further connected to sensors for closed-loop applications. To that aim, particular attention was ported on timing

considerations to keep channel-to-channel time coherence and to ensure hard real-time behaviour.

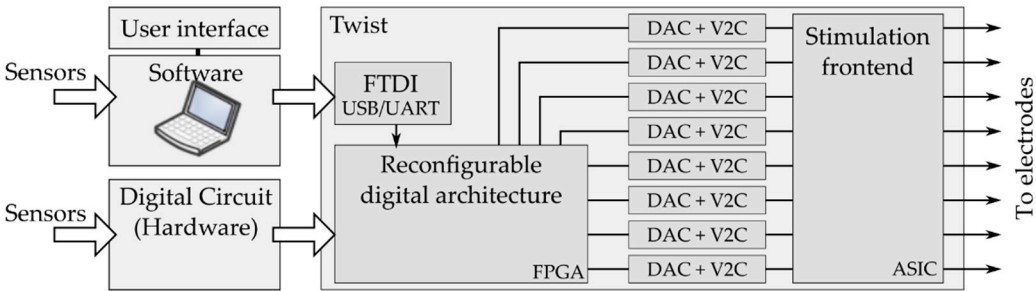

**Figure 3.** System-level architecture of TWIST: The system can drive up to 8 electrodes in parallel, the stimulation currents are high voltage compliant and generated by an ASIC. The ASIC receives its input from 8 parallel Digital to Analog converters (DAC) and associated voltage-to-current converter (V2C, low voltage). A dedicated digital architecture is implemented on an FPGA. This architecture can be controlled by other circuits or by a software user-interface, using a FTDI for USB connection.

### 2.2. Materials for Analog and Digital Designs

The integrated circuit was designed using a 0.35 μm HV CMOS technology from Austria Micro Systems (AMS) accessible through the Multi-Project-Center of GIP-CNFM (http://cnfmold.cime.grenoble-inp.fr/VersionUK/index_an.htm, 27 June 2021, design-kit ams H35B4D3, Cadence Design Systems). The electronic board was designed with National Instrument-Multisim Ultiboard 14.2. The reconfigurable architecture was described using VHDL inputs. The software interface was developed using Python. All code (VHDL and Python) and board designs are freely available on https://sourcesup.renater.fr/projects/opentwist (commit 'electronicsMDPI', 27 June 2021). The digital architecture was implemented on a commercially available FPGA board (Neso Board, Numato Labs Company, San Jose, CA, USA) to reduce development time and ease the replication of the system. This board contains the Xilinx Artix 7 FPGA (XC7A100T, Xilinx, San Jose, CA, USA), 2 Gb of DDR3 memory, a 128 Mb SPI Flash memory storing the FPGA configuration, a 100 MHz CMOS oscillator used as a system clock and a FTDI chip, in charge of the USB to UART bridge with up to 12 MBd performances. The FPGA board also contains all the necessary voltage regulators, allowing the board to be powered with a simple 5 V power supply. Finally, the Neso board gives access to 140 standard digital Inputs/Outputs.

### 2.3. Stimulation Front-End: A Versatile ASIC

#### 2.3.1. ASIC Architecture

Our primary objective was to produce a broad-spectrum stimulation system capable of driving different kinds of electrodes, which are the physical interface between a biological tissue and the electronic circuit. As the end-of-chain load, this element is crucial in the development of a stimulation circuit. Depending on its electrical impedance, the electrode determines the maximal voltage at the required current and pulse width. The type of electrode is selected by physiologists for a specific application, for mono- or multi-channel stimulation; its geometrical properties and constitutive materials depend on the targeted volume of triggered excitable cells. The geometry of an electrode has also a direct impact on its impedance as well as the surrounding tissue. Although it is demonstrated that the electrode impedance presents a complex behavior including non-linearity and fractional derivative [46], a simplified RC model is often used for electronic design, as in [37]: the capacitance is related to the metal-tissue interface and is directly proportional to the electrode area, while the extracellular medium can be modelled in first approximation by its resistivity. Macro-electrodes (having a pad area of more than a mm$^2$) have larger capacitance and lower resistance (C in the range of nF and R in the range of kΩ) than micro-electrodes (C in the range of pF and R from 10 kΩ to several MΩ).

We designed a modular architecture based on *Elementary Stimulation Channels* (ESCs). ESCs will be combined in various stimulation topologies, to suit a scheme of micro-electrode arrays with a high channel density and low-level currents, or one with few channels and high-level currents for larger macro-electrodes. To be compatible with configurations where multiple channels share a unique current return electrode, each ESC has a monopolar output with symmetrical power supply. This topology enables the asynchronous control of the ESCs, their combination, and active charge balancing.

Figure 4 illustrates different combinations of these ESCs. In the 'Stand-alone channels' mode, elementary ESCs are used independently; stimulation channels can be asynchronous, where each block is driven independently. In the 'Ganged Output channels' mode for larger electrode geometries, elementary blocks are combined, and outputs are summed on the electrode. The number of ganged blocks can vary with respect to the total current requirement. In the 'Current/field steering channels' mode, ganged blocks can also be combined in complex topologies capable of current steering over multi-polar electrodes, as carried out by [47]. In order to limit the number of IC pins, each fabricated ASIC comprises 8 ESCs.

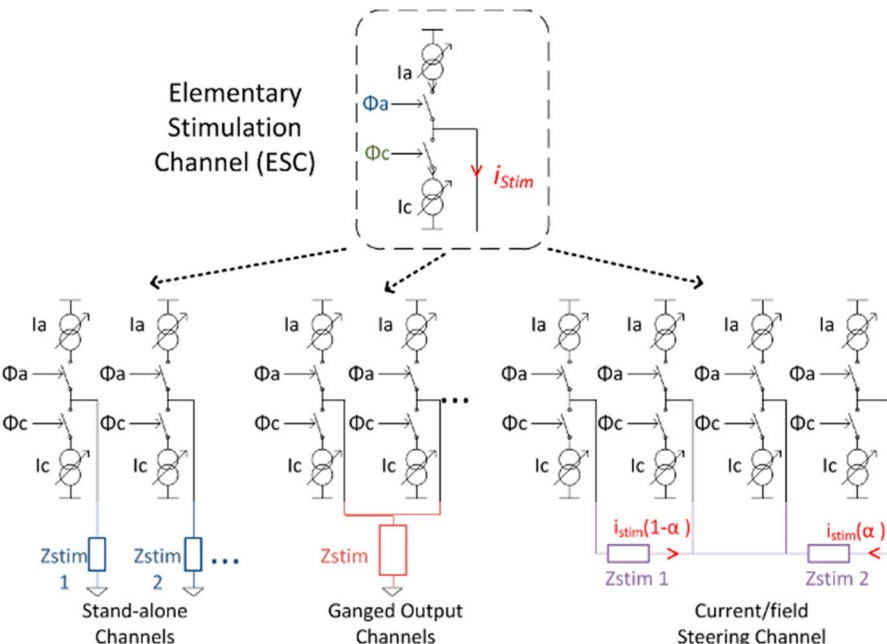

**Figure 4.** Illustration of the multi-application approach. One Elementary Stimulation Channel (ESC) includes a controlled bidirectional current source to stimulate the smallest electrode with a biphasic waveform. An electrode is represented by $Z_{stim}$, $i_{stim}$ is the stimulation current. *Ic* and *Ia* are current sources; switches are controlled by Fc and Fa. Several ESCs can be associated to provide multi-channel stimulation, stimulation of larger electrode or even complex stimulation schemes such as current steering with multi-polar electrodes; α represents the current steering ratio.

### 2.3.2. Elementary Stimulation Channels

An Elementary Stimulation Channel (ESC) has a current-mode analog input and digital inputs to control the cathodic and anodic commutations. These input signals are low voltage (LV) while the electrode output voltage can reach levels higher than the standard CMOS supply voltage. Therefore, we used a High-Voltage (HV) technology for the ASIC, namely the AMS H35 (0.35 μm) process that includes transistors dealing with up to 50 V. The maximum output current for one ESC has been fixed to 1 mA, allowing for a stimulation current up to 8 mA if all channels are ganged for a one channel macro-electrode. Figure 3 shows the schematic of a full ESC. It can be divided into three sub-circuits as described in Figure 5.

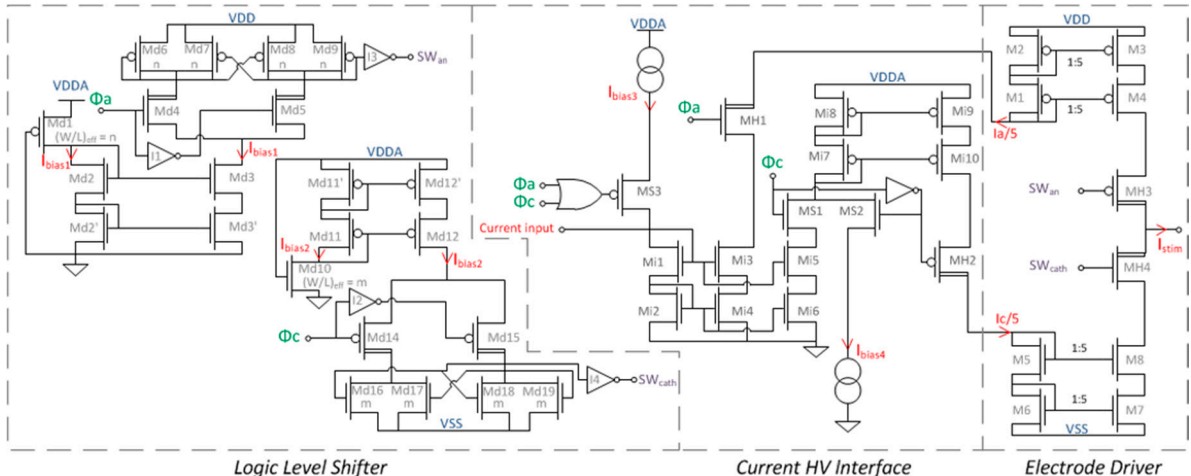

**Figure 5.** Schematic of the ESC circuit. From right to left, the ESC is composed of the electrode driver delivering the stimulation current to the electrode, an LV to HV interface block and a purely logic block combining the anodic and cathodic control signals. SWan and SWcath are internal signals; $I_{bias1}$, $I_{bias2}$, $I_{bias3}$, $I_{bias4}$ are internal biasing currents. Ic and Ia are current sources; switches are controlled by Φc and Φa.

(a)　Electrode Driver

This circuit (Figure 3, on the right) is connected to the electrode through an external DC blocking capacitor. To deliver the current stimulus $I_{stim}$ with the highest voltage compliance, it is supplied with high voltage VDD/VSS = ± 20 V. The Electrode Driver is composed of two current sources based on cascode current mirrors. The mirror formed by transistors M1 to M4 is used for the anodic current and the mirror M5 to M8 for the cathodic current. These two sources use LV transistors which have a higher gain factor and lower early effect than HV transistors; this optimizes the mirror accuracy and output impedance. Both anodic and cathodic current sources are designed with a current gain of 5.

Since the drain-source voltage of LV transistors cannot exceed 3.3 V, the current source voltage is controlled with two HV transistors acting as anodic and cathodic switches, MH3 and MH4. These HV transistors with thin gate oxide have approximately the same threshold voltage as LV transistors. The gate-source voltage of HV transistors cannot exceed 3.3 V, which limits the voltage of the current sources, M3–M4 and M7–M8, to less than 3.3 V. To reach the HV input voltage of this bi-directional current source, two interface blocks were designed: the Current HV Interface generates the anodic and cathodic currents with potentials shifted from LV to HV, and the Logic Level Shifter acts in the same way for the digital signals controlling the switches.

(b)　Current HV Interface

This circuit (Figure 3, center) takes as input an LV current source (labelled 'current input' on the schematic), the LV anodic and cathodic logic commands and generates the currents with HV compliance for the Electrode Driver. It is supplied by VDDA = 3.3 V.

In the event of an anodic stimulation, a current mirror with a gain of 1 formed by Mi1 to Mi4 is triggered by the anodic command Φa. This cascode current mirror performs a precise current copy and relies only on LV transistors. The anodic current is provided to M1–M2 through the HV transistor MH1. This transistor maximal drain-source voltage is larger than the HV supply (20 V) and ensures the voltage compliance. It also limits the power consumption when the anodic stimulation is off; it has a thin gate oxide and is directly controlled by the LV Φa command.

On the opposite, in the event of a cathodic stimulation, a first LV cascode current mirror with a gain of 1 formed by Mi1–Mi2 and Mi5–Mi6 is triggered by the cathodic command Φc; the current is then flipped with a second LV current mirror formed by Mi7 to Mi10. The cathodic current is provided to M4–M5 through the HV transistor MH2 ensuring the voltage compliance. In order to prevent pointless power consumption, the

transistor MS1 is directly controlled by Φc, which disables the branch formed by Mi5 to Mi8. A biasing current ($I_{bias4}$ = 0.8 μA) is added through MS1 and subtracted through MS2 to prevent LV transistor Mi10 from any drain voltage drift over its compliance. The same process is applied to Mi3 with the biasing current $I_{bias3}$ controlled through MS3.

(c)　　Logic Level Shifter

The Logic Level Shifter circuit (Figure 3, on the left) takes as inputs LV logic commands (0–3.3 V) of cathodic or anodic stimulation and generates the signals controlling the switches MH3 and MH4 of the Electrode Driver, respectively SW$_{an}$ and SW$_{cath}$. The transistor MH3 requires voltage levels from VDD (VDD = 20 V) to (VDD = 3.3 V). The transistor MH4 requires voltage levels from VSS (VSS = −20 V) to (VSS + 3.3 V). This circuit is powered by both LV (VDDA) and HV (VSS and VDD) supplies. The two sub-circuits for anodic and cathodic circuits are symmetrical and only the anodic level shifter will be described further.

The anodic Logic Level Shifter relies on a voltage mirror formed by LV transistors Md1 and Md6 to Md9. Md1 is used as a static current generator providing *Ibias1* with a corresponding W/L = n. The n-ratio is applied to two pairs of n-sized transistors, Md6–Md7 and Md8–Md9. These two pairs are controlled through the two HV switches Md4 and Md5, controlled by the Φa anodic command and its complement. The current $I_{bias1}$ passes through Md6–Md7 or Md8–Md9, which are connected to the HV supply VDD. It generates a potential difference of 3.3 V across the concerned pair attached on the High voltage level VDD. An additional inverter I3 ensures the buffering of the shifted logic signal SW$_{an}$.

## 2.4. Digital to Analog Conversion Stage

A dedicated D/A stage is implemented to control the stimulation channels (see Figure 6). It includes one serial 25 Msps DAC per channel (Analog Devices AD5424) that enables us to drive up to 8 channels simultaneously and independently. The DAC voltage reference can be adjusted to set up the conversion gain. All DAC outputs are followed by a precision voltage to current converter, as represented in Figure 6. The converter's output is mirrored via a cascode current mirror, to provide a low voltage high-impedance output, and is then fed into the ASIC's inputs.

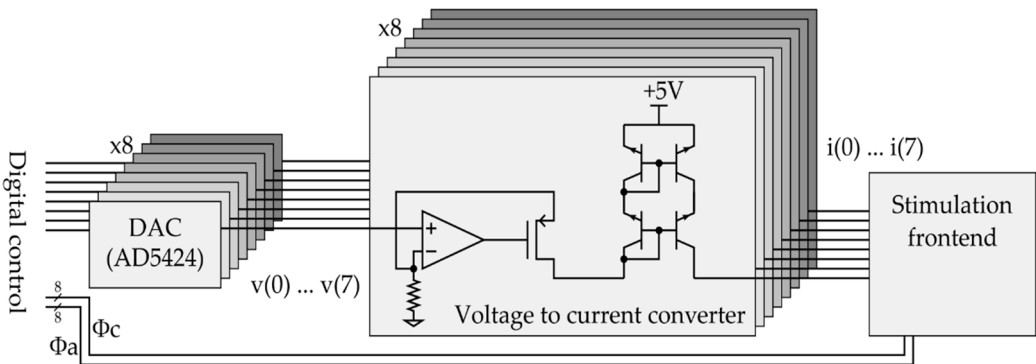

**Figure 6.** D/A stage (DAC followed by Voltage to Current Converter) implemented on TWIST with off-the shelf components.

In order to facilitate the system usage in various experimental conditions, we choose to power the whole system with a simple 5 V power supply. Thus, the system is equipped with voltage converters and regulators to produce the required supply voltages. The symmetric high voltages are obtained using boost/inverting converters (LT8582, Linear Technology, Milpitas, CA, USA). Their outputs are fed into linear voltage regulators to provide clean DC voltages for the stimulation ASIC (VSS/VDD = ±20 V, VDDA = 3.3 V and VSSH/VDDL = ±16.7 V). To improve the system reliability, a low-cost 8-bit MCU (MC9S08, NXP Semiconductors, Eindhoven, Netherlands) manages the start-up sequence of the power supplies. Finally, TWIST provides 4 PMOD (Peripheral Module Interface)-

compatible I/Os connected to the FPGA, which can be used for external communication, input or output triggering, or to add hardware extensions.

### 2.5. Reconfigurable Digital Architecture

#### 2.5.1. Overview of the System Architecture

The TWIST stimulation system runs as a slave obeying orders from an experiment master (computer/microcontroller or specific hardware). It handles high level decisions on the experiment, such as stimulation start/stop, waveform definition, waveform modulation or trigger generation. All the stimulation control circuits are implemented in the FPGA. They consist in:

- Waveform generators (WG): deliver the output waveforms provided to the DACs;
- Analog channel controller (ACC): manages analog output channels;
- Trigger management (TrM): handles synchronization and WG start;
- Instruction decoder: provides the logical interface with the master;
- Control interface: manages the physical interface with the master.

The overall architecture is drawn in Figure 7. The number of ACCs (Analog channel controllers) is directly related to the number of analog outputs of the hardware system (1 ACC per ASIC ESC). The number of WGs depends on the application and may widely change from an application to another. The number and variety of WGs is defined at synthesis level.

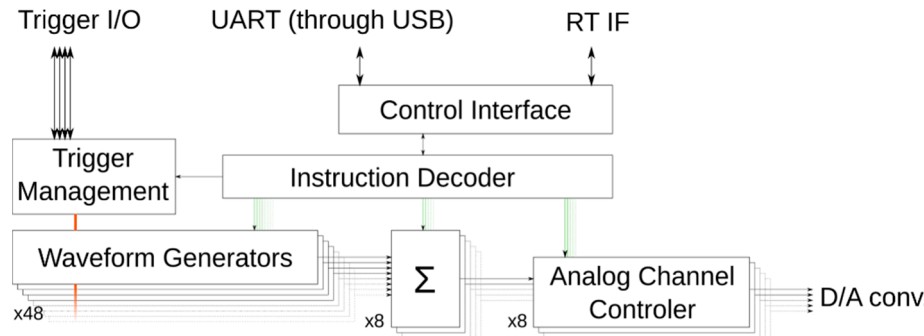

**Figure 7.** Description of digital architecture. Stimulation commands are received by the Control Interface and the Instruction Decoder through a USB/UART bridge or through a proprietary Real-Time interface (RT IF). The Instruction Decoder controls all Waveform Generators and manages the adders to feed the eight Analog Channel Controllers that are the direct interface to the Digital to Analog converters. Stimulations can be initiated by triggers and all waveforms are synchronized by the Trigger Management unit.

Data produced by the Waveform Generators (WGs) are coded with 16-bit signed representation. Each of the WG can contribute to any of the outputs, following a combination pattern described in Figure 8. For each ACC, a programmable mask determines whether the WG is contributing to build the output signal. This masking is symbolized by switches in Figure 8. All contributions are then added to build the final output waveform. Due to the potential high number of WGs, a pipelined adder tree has been implemented to mix any combination of WGs with a fixed output latency. In the current controller implementation, the adder can combine up to 64 WGs in 6 clock cycles. As the current implementation contains 48 WGs, a synchronous register was placed to balance propagation delays and ensure proper synchronicity among waveforms (strictly identic delay from signal generation and data output). The consequent latency of the full addition is 50 ns, or one sample at 20 MHz sampling.

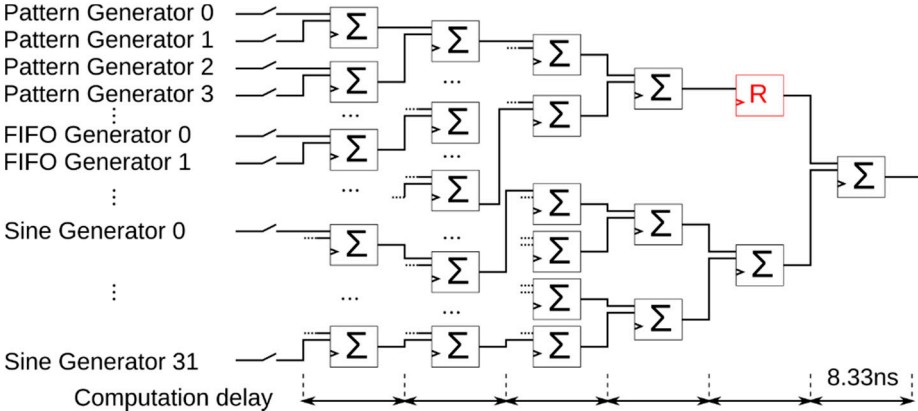

**Figure 8.** Architecture of the adder tree combining the WGs outputs. In this example the WGs are a series of Pattern Generators, FIFO Generators and Sine Generators. The simple register *R* marked in red is present for the unique purpose of balancing computation delays and keeping time coherence among added signals, whatever the original WG is. The total computation delay is 50 ns (8.33 ns for each single adder).

ACC is the only hardware-specific module of the hierarchy. It controls the digital to analog converter (DAC), as well as the anodic and cathodic CEs (channel enable). All ACCs of the system are fully synchronized to ensure coherence between the signals produced on all the system outputs. The output sampling frequency is programmable from 1 MHz to 20 MHz by steps of 1 MHz. This implementation choice comes from the necessity to provide a strictly deterministic response to each trigger (sampled at 1 MHz. Therefore, only a subset of the output frequencies is rigorously periodic (1 MHz, 2 MHz, 3 MHz, 4 MHz, 5 MHz, 6 MHz, 8 MHz, 10 MHz, 12 MHz, 15 MHz and 20 MHz). The high 20 MHz limit is determined by the DAC.

The system provides 8 unipolar outputs able to reproduce accurate pulsed waveforms. However, linearity issues at sign change may occur and decrease the quality of waveform rendering. To improve waveform accuracy, ACCs are pairable. In normal mode, each ACC receives its own waveform to produce and output it. In paired mode, the even numbered ACCs only outputs the positive part of the waveform it receives, while the odd numbered ACC outputs the negative part of the waveform of the paired even numbered waveform. Both odd and even ACCs also produce a tunable offset to keep a minimal output current (positive for even, negative for odd); these offsets keep outputs in their best linearity zone and compensate each other.

ACCs also provide an inversion function to output a current opposite to the value issued from the generators. The rigorously synchronized output structures then make it possible to perform a bipolar stimulation.

### 2.5.2. Waveform Generators

The system described in Figure 8 currently contains three types of waveform generators:

- Pattern players: These address conventional pattern-based stimulators. The master uploads a waveform to the TWIST board, which is played as soon as the associated trigger is active at a maximal rate of 20 Msamples/s. The default system contains 8 pattern players, each of them associated to a 16 k-sample memory. Although the system architecture does not restrict the maximal number of pattern generators, the instruction set is able to address only 16 of them. This limitation was considered secondary since the system only provides 8 analog outputs.
- FIFO outputs: These are intended to provide a continuous stream of data issued by the master. These generators are different from the pattern player in that their associated memory is not dedicated to store a repeated pattern, but only as a buffer to maintain the stability of the data flow. Each FIFO output can produce data at a

rate of 20 Msamples/s; however, in the default system, communication limits the maximum sample rate to 5 Msamples/s distributed on 8 FIFO outputs. Although these generators are designed for continuous streaming, they are trigger-sensitive to ensure synchronous streaming and possibly synchronous stimulation on parallel channels.

- Sine generator: This provides sine waveforms computed internally. The output frequency is programmable from 1 mHz to 1.19 MHz by steps of 1 mHz. As there are 32 sine generators on the default system, the generator architecture has been chosen to be both fast and low resource demanding. Sine values are usually computed using a CORDIC architecture [48], but this technique suffers from large hardware resources requirement and computation latency. We chose to implement sine wave computation using a recursive generator based on the forward Euler resolution of the equation f″(x) = −f(x). The integration step is a multiple by a power of 2 of the digital clock, so that no multiplier is necessary but only bit shifting operations. When the desired frequency is not directly achievable by a multiple of a power of 2, the nearest candidates are used alternatively during the generated sine period so that the average of the integration step corresponds to the targeted frequency.

As shown in Figure 8, the architecture contains 8 Pattern players, 8 FIFO outputs and 32 sine generators (for a total of 48 pattern generators).

### 2.5.3. Interfacing and Control

As stated in Section 2.1, the TWIST system architecture is defined to be controlled by a master which actually has an overview on the experiment. The system has two interfaces:

- The first interface is dedicated to computer-based control. It provides UART connectivity through USB connection using a *FTDI* bridge. A control library has been developed using python3 language. This interface runs at 921,600 bps (bits per second) and is suitable for high level control of the stimulation (load stimulus waveform when stimulation is off, manual start and stop).
- The second interface is dedicated for real-time hardware control. It proposes 3 interface modes: standard UART, high-performance and SPI. The standard UART mode makes sure that any digital hardware device can control the system. It supports communication speed from 9600 bps to 33.3 Mbps by steps of 200 bps. The high-performance mode is based on three 16-bit words UART lines from master to TWIST and one feedback line from TWIST to the master. The three combined lines make transfers up to 100 Mbps possible, while 16-bit words improve communication efficiency. The SPI mode is finally available for synchronous transfers. It handles 40 Mbps bidirectional transfers without hardware protocol overload. This interface requires 4 I/Os. A detection mechanism lets the system determine which mode is used (UART, SPI or High performance) so that any controller with standard UART or SPI works without specific configuration.

For both computer and real-time interfaces, the system latency is 30 ns; therefore, the actual experiment latency is defined by data transfers and master intrinsic latency. Both interfaces use the same instruction set, based on 16-bit words. Each instruction is followed by a confirmation code, but as all communication schemes are fully duplexed, instructions can be pipelined without protocol over cost.

## 3. Results

### 3.1. Hardware Electrical Characterization

### 3.1.1. Chip Fabrication and Characterization

The chip for current stimulation was fabricated using AMS HV CMOS H35B4D3 process (AMS, Premstaetten, Austria). The chip is presented in Figure 9c. The core area is 3.9 mm$^2$. However, a large area is dedicated to the pad ring, resulting in a total die area of 4.78 mm$^2$. The ring is divided into two sections; HV pads are only used for high voltage power supplies (+/−20 V and +/−17.6 V) and the eight ESC outputs, on the left of the

microphotography (Figure 9). A low-power pad ring is used to interface the circuit with other external components. The chip can be encapsulated into QFP100 packages.

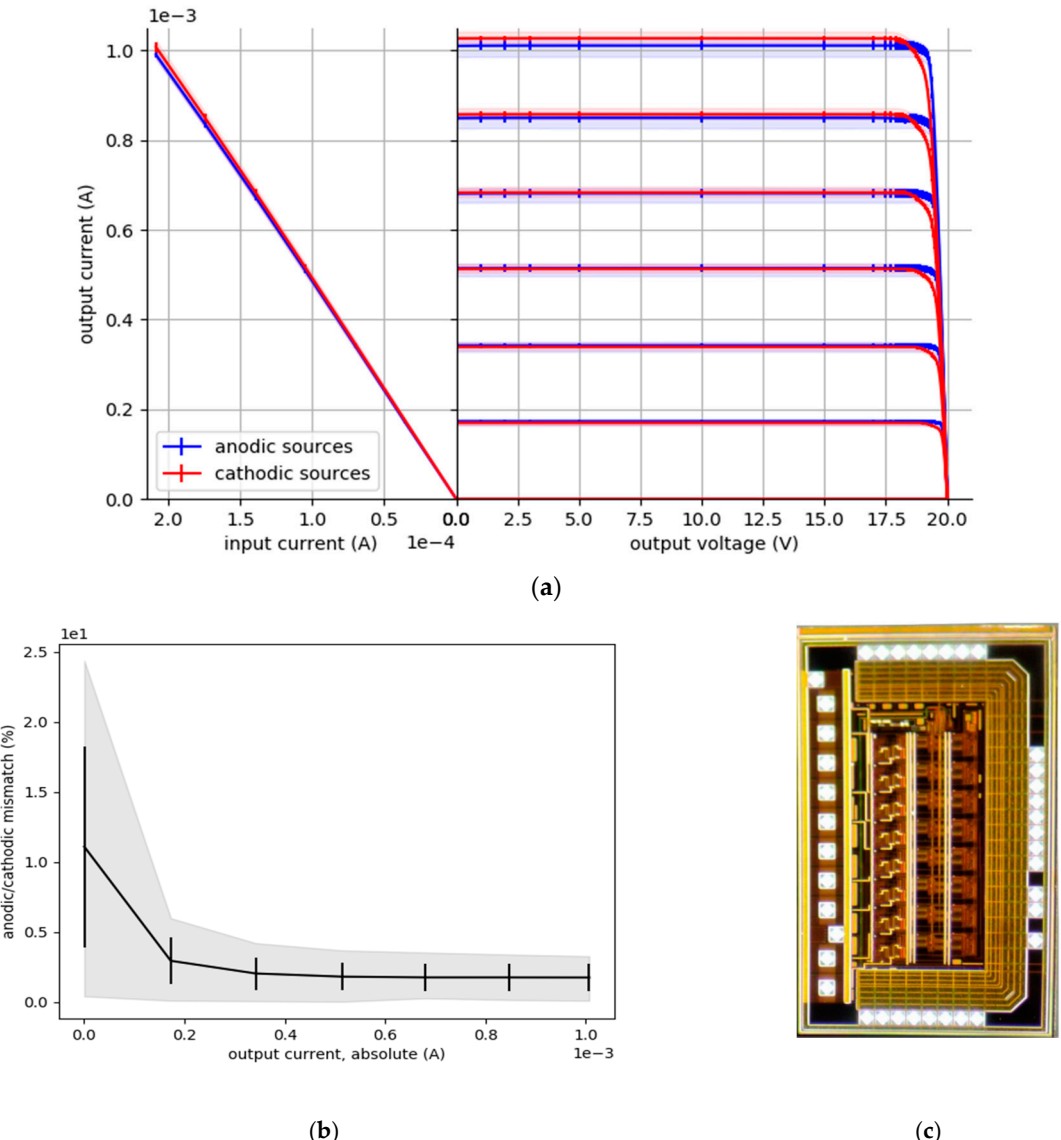

**(a)**

**(b)**　　　　　　　　　　　　　　　　　　　**(c)**

**Figure 9.** (**a**) Elementary Stimulation Channels characterization measurements; both transfer (output current as a function of input current) and output (output current as a function of output voltage) characteristics for anodic and cathodic sources are plotted. Curves represent the average characteristic, bars the standard deviations, and colored areas the maximum range. (**b**) Fine characterization of anodic versus cathodic mismatch over the stimulation current range. (**c**) Microphotography of the chip SONIC for High Voltage current stimulation of electrodes; HV pads are located on the left of the chip, LV pads on top, right and bottom sides. The die chip is 4.78 mm$^2$ large and core-limited.

ESCs have been firstly electrically characterized by their static performances. Measurements were performed on three chips (3 × 8 = 24 individual channels). Anodic and cathodic current sources have been characterized using a Keithley 6487 (Keithley, Cleveland, OH, USA) Instruments picoammeter and voltage source. The transfer and output curves are plotted in Figure 9a and the resulting static characteristics are summed up in Table 1. The current gain is slightly under the targeted gain of 5 (see schematic on Figure 4), this gain can be corrected using the digital control circuitry. The gain dispersion remains under the percent. A mismatch between NMOS and PMOS HV transistors has for consequence a difference between anodic and cathodic gains. The equivalent current mismatch has been computed and plotted in Figure 9b. This mismatch is high for low-current values and

remains in the range of 2% for currents higher than 300 μA, which is the typical stimulus range with small micro electrodes.

**Table 1.** ECS static characteristic (Average values, σ for the standard deviation when given. Statistics on 24 measured channels. Worst case is the minimal measured value for current sources output impedances).

|  | Anodic | Cathodic |
|---|---|---|
| Current Gain | 4.77 ($\sigma = 2.18 \times 10^{-2}$) | 4.85 ($\sigma = 4.19 \times 10^{-2}$) |
| Voltage Compliance | 18.3 V | 17.9 V |
| Output Impedance | 5.92 MΩ | 10.1 MΩ |
|  | (Worst case:1.87 MΩ) | (Worst case: 1.66 MΩ) |

The output impedance has been computed from the output characteristics and is listed in Table 1. This value is usually considered in the literature to determine the ability of a stimulator to drive an electrode considering its impedance. The minimal voltage compliance is 17.9 V in standalone channel, 36.2 V in differential configuration with two ECS with polarity inversion at each electrode pole. These results correspond to the average state of the art for conventional stimulators described in Section 1 (points of the Figure 2a).

### 3.1.2. Static Linearity

The system linearity has been evaluated by considering the stimulation system (ASIC and digital architecture) as a digital to analog converter. The DAC used to drive the ASIC has an 8-bit resolution, and the ASIC adds a sign bit. Input code varies from −255 to 255; the corresponding LSB on a stimulation channel is about 3.9 μA. The linearity was evaluated on a resistive load of 10 kΩ by measuring the DC current level on the load. Results are plotted in Figure 10.

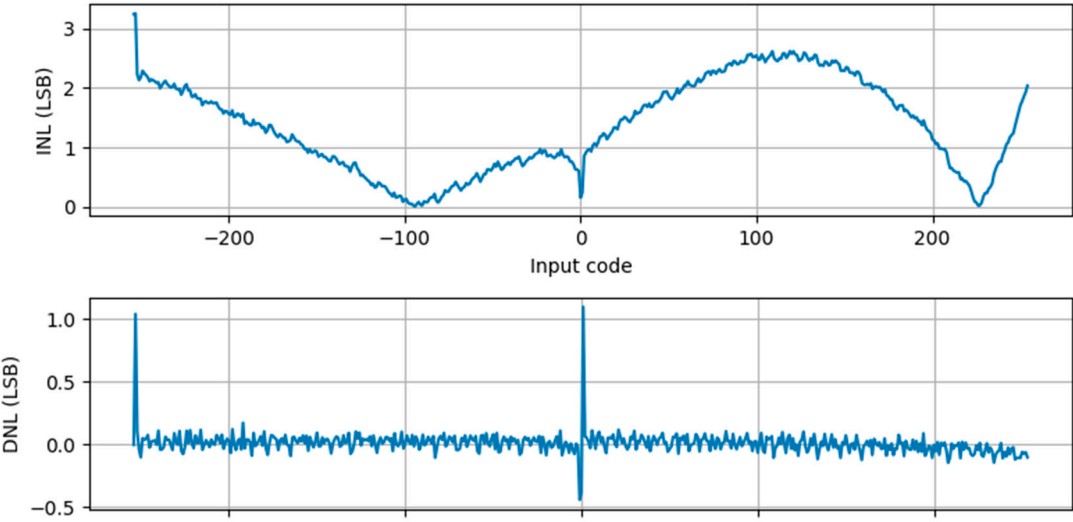

**Figure 10.** Static linearity of the stimulation system: the measured quantity is the current over a 10 kΩ load. Top curve: Integral Nonlinearity (INL). Bottom curve: Differential Nonlinearity (DNL). INL and DNL are expressed in LSB = 1 mA/256 ≈ 3.9 μA.

The integral linearity remains below 3 LSB. The maximum amplitude for a code and its opposite (corresponding to the case of a conventional stimulation with opposite current for charge balancing) gives an unbalanced residual error of 2.6 LSB corresponding to a maximal current balancing deviation of 10.1 μA (balancing error of 2.6%). This error is systematic and can be corrected at a higher level, potentially at the software level. The

differential non-linearity remains low, on average under 0.1 LSB. It ensures a monotonic stimulation current, except on the lowest code and at the 0 crossing.

### 3.1.3. Frequency/Load Stimulation Range

In this part, we evaluate the ability of the stimulation system (ASIC and digital architecture) to provide a precisely controlled signal. We define the gain in current as the ratio between the current amplitude on a load divided by the programmed current amplitude. This gain can be measured over a frequency range to perform Bode plots; however, the gain is also affected by the load. An evaluation of the gain has been performed from 100 Hz to 1 MHz, from 100 Ω to 1 MΩ. For instance, with a 1 MΩ load, with the voltage compliance of the ASIC, the maximum peak to peak amplitude of a signal is of 4.6 LSB for a single ESC, and 9.2 LSB for two ESC in differential mode. Results are shown in Figure 11a. For comparison purposes, we performed electrode impedance spectroscopy measurements on different electrodes and superimposed obtained curves (CUFF: MicroProbes NC-2.5-3-125umSS, DBS: electrode design used in [49], LIFE: electrode design of [50], MEA, Multichannel Systems MEA 100/10-Ti-gr, TIME: Neuronexus probe, Cardiac: Medtronic 5554 Capsure Z Novus lead). All spectra have been measured on electrodes in the same saline solution, so that these curves highlight geometric properties of the electrode kind.

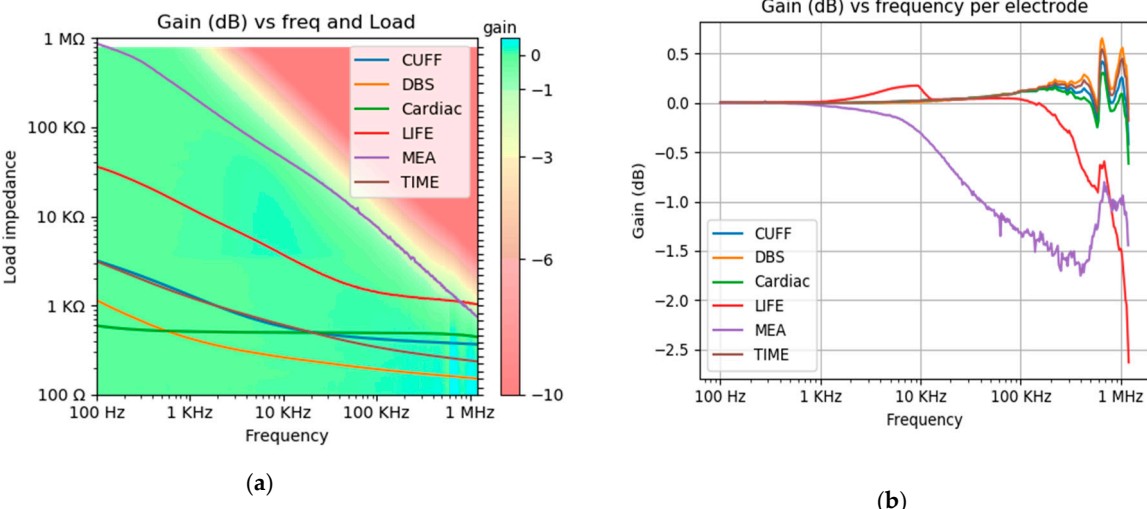

(**a**)

(**b**)

**Figure 11.** (**a**) Current gain between the generated current measured on a resistive load and the targeted current amplitude programmed on the stimulation system, versus frequency and load. The gain range is color-coded. Ticks on the right y-axis indicate the load values used to compute the colored surfaces. Impedance spectroscopic measurements performed on different types of electrodes are superimposed (color curves). (**b**) Gain versus frequency for each electrode, based on electrode impedance spectroscopy curves from (**a**).

In Figure 11a, the green zone corresponds to an attenuation of less than 1dB. This area has three boundaries:

- An upper load limit (maximum load limit), resulting from the voltage induced by current quantum (output DAC LSB) compared to the maximum compliance voltage. This limits the highest possible load at low frequencies. Pushing this boundary implies either increasing the number of bits of the digital to analog conversion or increasing the voltage compliance.
- An upper frequency (maximum frequency limit) limit related to the sampling frequency of the digital to analog conversion. The 20 MHz output sampling bounds the theoretical limit to 10 MHz, but Figure 11a stops at 1 MHz to preserve sine waveforms fidelity and avoid potential misinterpretation.

- A boundary with the yellow part of the gain scale (−3 dB limit), induced by the dynamic output parasitic impedance of the ESC. This boundary (minimum gain limit) follows a 1 decade per decade trend, related to the parallel resistive-capacitive nature of the parasitic impedance of the output current mirrors or the ESC. Pushing this boundary implies to reduce the size of output channel transistors.

Interestingly enough, all electrode impedance spectra remain in a zone of low attenuation. Figure 11b shows the exact value of the gain for all electrode impedance depending on the frequency in Figure 11a. All attenuations are less than 3 dB, even on the smallest electrodes (MEA) corresponding to the highest possible impedance.

### 3.1.4. Dynamic Linearity

Taking into account the potential therapeutic impact of complex and high frequency stimulation waveforms, we studied the dynamic linearity of the system as a function of the frequency, by evaluating the signal to noise and distortion ratio (SINAD) of the stimulating device. As the digital to analog conversion is performed using an 8-bit DAC, the maximum SINAD is theoretically limited to 50 dB approximately. The SINAD is expressed as the ratio of the RMS value of the full signal and the RMS value of the noise and distortion/spurious tones. RMS values were computed in the frequency domain, using a notch filter to filter out the frequency of interest. Figure 12 shows the measurement results. The equivalent number of bits (ENOB) levels are indicated for comparison purposes.

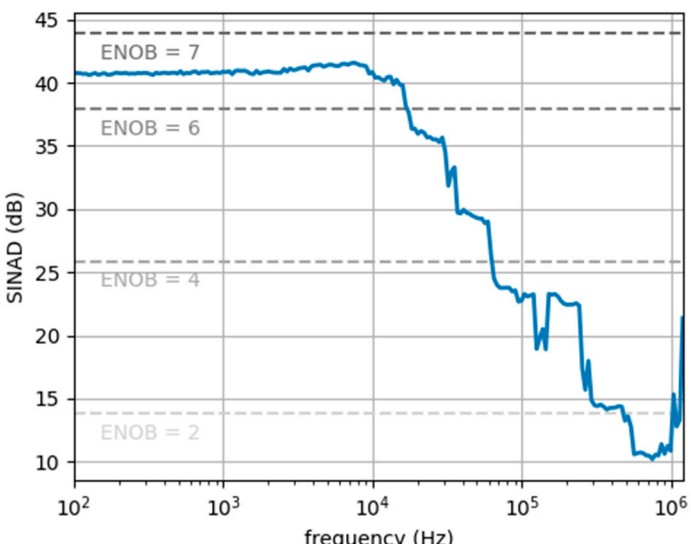

**Figure 12.** SINAD evaluated by Total Harmonic Distortion plus Noise (THD + N) computation. Equivalent number of bits (ENOB) levels are plotted to highlight the performances of the stimulation system when considered as a converter over the frequency range of interest.

From 100 Hz to 10 kHz, the SINAD is about 40 dB (ENOB around 6.5 bits), which can be explained by the 8 bits DAC (practical ENOB between 7 and 8 bits) and the additional noise from the other circuits (ASIC and voltage to current converters). SINAD also takes distortions and spurious tones into account, explaining the lower value obtained compared to the theoretical value. From 10 kHz to 1 MHz, we observe a linear decrease of the SINAD, which can be explained by the way sine waves are digitally computed. Indeed, internal numbers of bits during computation are reduced with sine wave frequency, in order to maintain a reasonable amount of digital resources required. For example, after 600 kHz the waveform is only computed using 3 bits, resulting in a poor SINAD value.

### 3.1.5. Digital Resource Consumption

The digital architecture hardware requirements are detailed in Table 2. All parts are detailed at a function level to show the absolute and relative cost of each function. However,

adding or removing channels or generators would also have an impact on the interface part, in which resources are difficult to identify at the function scale. We chose to display the resources of each function as if it was implemented in a single instance, but the total resources may be reduced because of shared resources and architecture-scale optimization.

**Table 2.** Hardware resources of the architecture elements. RAM resources are expressed in kbits instead of blocks to facilitate comparison. We consider that device block RAMs are 32 kb instead of 36 kb, to reflect the actual percentage of blocks used. All percentages are architecture related, except for the final table line, where percentages are device-related.

|  |  | LUT (%) | Flip Flops (%) | RAM (%) |
|---|---|---|---|---|
| Interface/Control |  | 4507 (15%) | 5424 (21%) | 32 kb (<1%) |
| Pattern generator | total | 1139 (4%) | 424 (<2%) | 2 Mb (50%) |
|  | single | 383 (<2%) | 53 (<1%) | 256 kb (6%) |
| FIFO generator | total | 1800 (6%) | 712 (<3%) | 2 Mb (50%) |
|  | single | 435 (<2%) | 89 (<1%) | 256 kb (6%) |
| Sine generator | total | 16,423 (55%) | 8864 (34%) | 0 (0%) |
|  | single | 724 (<3%) | 277 (1%) | 0 (0%) |
| Final adder | total | 5184 (17%) | 9728 (37%) | 0 (0%) |
|  | per channel | 648 (<3%) | 1216 (5%) | 0 (0%) |
| Channel control | total | 1057 (4%) | 1104 (4%) | 0 (0%) |
|  | per channel | 134 (<1%) | 138 (<1%) | 0 (0%) |
| Total |  | 29,982 (100%) | 26,261 (100%) | 4128 kb (100%) |
| Device (xc7a100t) |  | 63,400 (47%) | 126,800 (21%) | 4320 kb (96%) |

Although the design computes heavy mathematical functions, no DSPs are used, leaving room to implement higher level features.

Sine generators represent a major part of the architecture. However, other implementation strategies require even more resources. We actually considered using a CORDIC-based architecture. It required 568 6-input LUTs and 512 Flip-Flops for the simple sine waveform generation, not to mention at least two DSP per generator for time and amplitude scaling, and additional logic for their configuration.

Finally, no PLL usage is displayed. However, because of the embedded 100 MHz oscillator, the design requires one PLL block to generate the local 120 MHz system clock. We do not consider this need as critical, because it is hardware-dependent. Should such a resource be critical, designing a specific board with a 120 MHz oscillator is achievable by any up-to-date designer.

### 3.2. Waveform Generation

While the ability of this system to drive different groups of electrodes is highlighted in Section 3.1.3, we also assessed the system capacity to successfully generate different waveforms. As explained in the introductory section, many stimulation paradigms require going beyond the simple biphasic waveshape. In Figure 13, we provide current and voltage measurement of waveshapes produced by our system:

- Biphasic waveforms and bursts: Figure 13a biphasic current with an amplitude sweep on a DBS electrode; Figure 14a,b with a burst pattern on a LIFE electrode.
- Energy efficient waveforms: Figure 13b with anodic pulse current waveforms as discussed in [20], on a TIME electrode.
- Increased selectivity waveforms: Figure 14c with the slowly rising pulse waveform discussed in [23] on a CUFF electrode, Figure 13d with the pre-pulse strategy from [22] on a TIME electrode.

- High-frequency blocking stimulation: Figure 14e,f with sinus stimulation at 1 kHz and 200 kHz, respectively, on a CUFF electrode.
- Biomimetic/controllable burst stimulation: Figure 15a,b with a gaussian modulated burst as discussed in [31] on a TIME electrode.

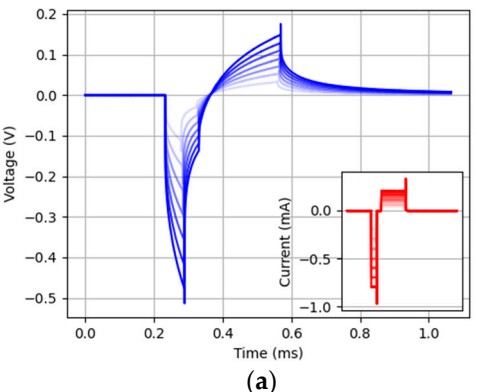

(**a**)

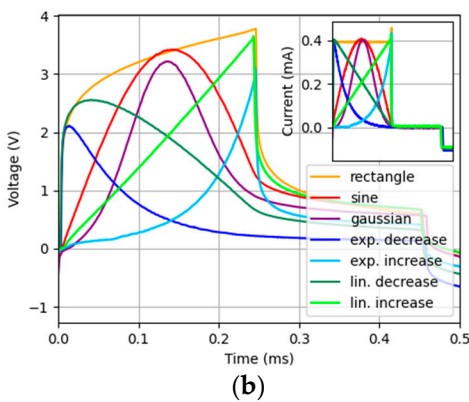

(**b**)

**Figure 13.** Stimulation waveforms as delivered on electrodes in a saline solution. Currents are recorded during the stimulation with a transimpedance amplifier; voltages are recorded on the electrode. (**a**) Typical DBS biphasic waveform (*Tc* = 60 µs, inter-pulse of 60 µs, active balancing with a current ratio of 1:5), from *Ic* = 100 µA to 800 µA with a 100 µA step, performed on a DBS electrode. (**b**) Current stimulation pulse, with different shapes as performed in [20] for energy efficiency optimization. All waveforms have an amplitude of 400 µA, are balanced with a second negative constant current pulse and have been applied on a TIME electrode.

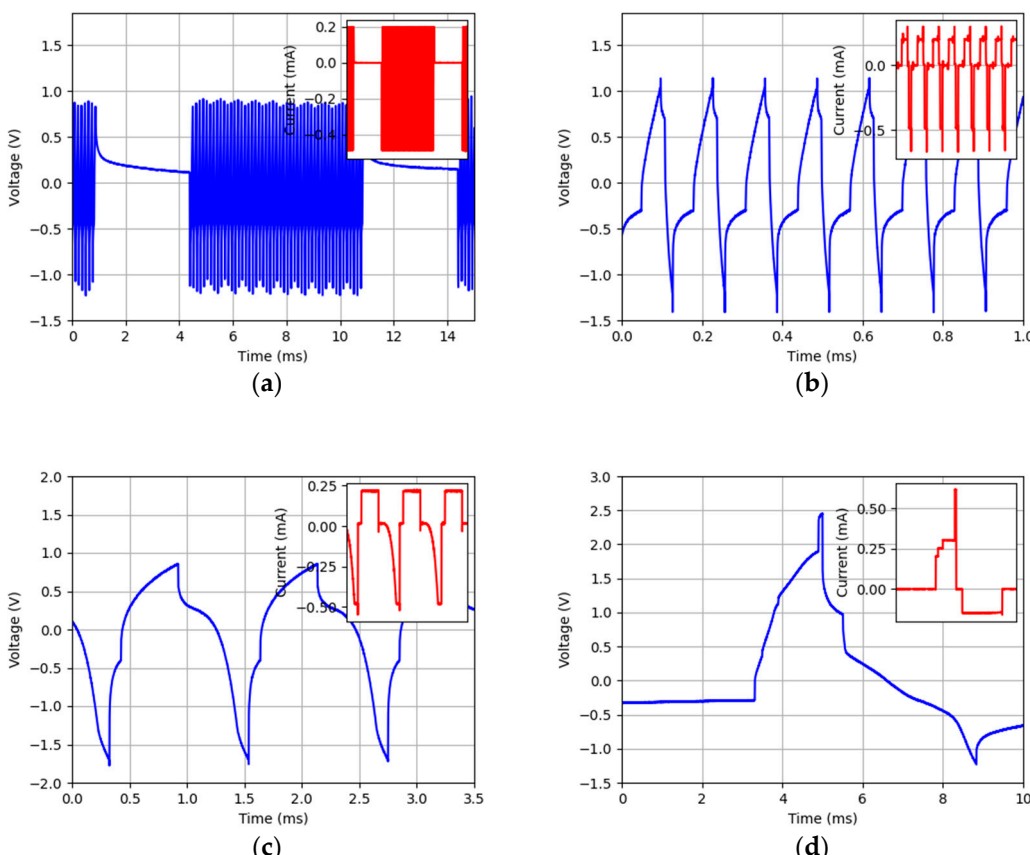

**Figure 14.** *Cont.*

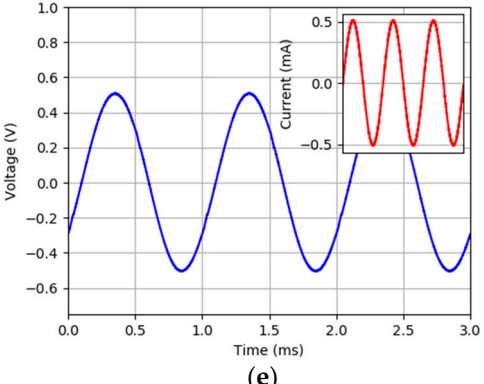
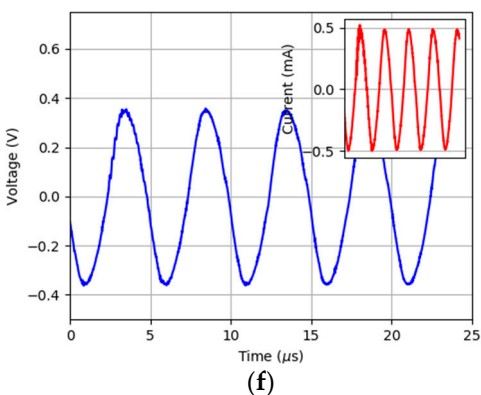

(e)  (f)

**Figure 14.** Stimulation waveforms as delivered on electrodes in a saline solution. Currents are recorded during the stimulation with a transimpedance amplifier; voltages are recorded on the electrode. All currents are plotted in red in insets; all voltages are plotted in blue. Insets have the same time windows as their parent plot. (**a**) Typical burst of biphasic pulses (*Tc* = 30 μs, inter-pulse of 10 μs, active balancing with a current ratio of 1:2, stimulation frequency of 8 kHz) applied on a LIFE electrode. (**b**) Temporal zoom on the same stimulus as (c). (**c**) Slowly rising pulse as performed in [23] for improved selectivity stimulation. The amplitude is 500 μA, the waveform has been balanced with a constant current anodic pulse. The waveform has been applied on a CUFF electrode. (**d**) Biphasic pulse with pre-pulse addition as described in [22] for cell-type selectivity. The waveform has been applied on a TIME electrode. (**e**) Sinusoidal waveform corresponding to High-Frequency Blocking Stimulation as performed in [26] with a frequency of 1 kHz and an amplitude of 500 μA applied on a CUFF electrode. (**f**) Sinusoidal waveform with a frequency of 200 kHz and an amplitude of 500 μA applied on a CUFF electrode.

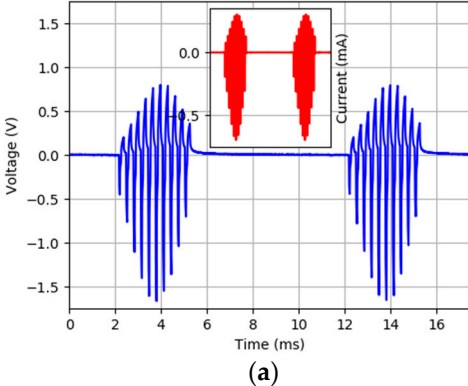
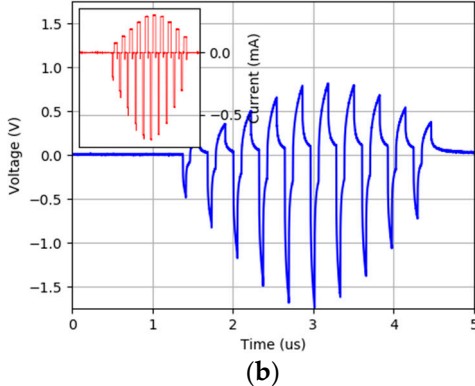

(a)  (b)

**Figure 15.** Stimulation waveforms as delivered on electrodes in a saline solution. Currents are recorded during the stimulation with a transimpedance amplifier; voltages are recorded on the electrode All currents are plotted in red in insets; all voltages are plotted in blue. Insets have the same time windows as their parent plot. (**a**) Controllable burst stimulation as performed in [31] consisting in a biphasic waveform (*Tc* = 60 μs, *Ta* = 180 μs) with a gaussian envelope applied on a TIME electrode. (**b**) Temporal zoom on the same stimulus as (i). All electrodes are fully specified in Section 3.1.3.

Further details on the waveshapes' parameters are given in the captions of Figures 13–15.

All waveforms were successfully generated. Glitches observed on some waveshapes are related to the stimulation switches opening on potentially parasitic inductive loads due to electrode wiring to the stimulation system. Moreover, as discussed in Section 3.1.4, the generation of high frequency signals is associated with distortion, as one can note on Figure 14f.

## 4. Discussion

The use of unconventional waveform stimuli, with well-defined spectrum (from few Hz to few hundreds of kHz), is largely supported by computational studies, while the most addressed features in conventional stimulators remain charge balancing, source

matching and power consumption. Although under increasing investigation, electronic stimulators capable of delivering unconventional waveforms are still poorly developed. In this contribution, we propose a complete Neuro-stimulation Environment for Arbitrary waveform generation. We have demonstrated its ability to generate all unconventional waveforms found in the computational studies, while loaded on most commonly used electrodes. Most interestingly, these performances are reached while subcircuits of the proposed architecture could still be further optimized.

The ability to properly generate unconventional waveforms—such a HF sinusoid, exponential decays or amplitude modulated burst—depends on the performance balancing of the overall stimulation system more than the individual optimization of subcircuits. The major constraint we identify is the capacity of the digital architecture to handle the data processing rate associated with the frequential content of the stimulus. A limit in our system was the ability for the FPGA to host the architecture for eight channels, which was assessed with different generator configurations. The system technology related figures on generating a well-defined spectrum are highlighted in Figure 11: at higher frequencies, our system is constrained by the minimum gain limit related to the ASIC technology, by the frequency limit imposed by the DAC and by the digital implementation strategy. While raising frequency over 10 MHz does not appear relevant due to the unavoidable electrodes parasitic capacitance, it would be interesting to reach a 0 dB gain for higher loads. This can be achieved with a second generation of ASIC, either by choosing a more advanced technology or by increasing the analog front-end output impedance. The output impedance of the ESC current sources could be maximized by changing the current mirror as in [35]. The anodic and cathodic gain mismatch could also be further reduced as in [34]. Other parameters such as dynamic linearity could also be optimized to increase stimulation efficiency, especially at higher frequencies [51].

At this stage of our study, power consumption figures are not a priority since we do not target implanted stimulators. Yet, we have not performed power consumption measurement and have only approximate figures, which could in a large part be improved if our digital architecture was implemented in a digital ASIC. In any case, power consumption will be the major limitation or challenge for the design of implantable systems capable of unconventional stimulation.

Our system offers maximum versatility on the stimulus waveform. Still, we are fully aware that the design of an implantable device will be specific to a therapeutic strategy and the associated stimulation pattern and electrodes. Hence, the characterization results in Table 2 should be considered as achievable targets and guidelines for the design of application-specific stimulators. We discuss below some strategies to optimize digital resources in the case of a more application-specific (thus less constrained) stimulator.

Our architecture is based on three different wave generators (Pattern, FIFO, Sine generators) that can be mathematically combined through adders. Both Pattern and FIFO generators are RAM-consuming, and the size of the memory is directly correlated with the complexity of the targeted signal. In our proof-of-concept system, RAM resources allocation was driven by its availability on the FPGA (which explains its high number). The most resource consuming modules pointed in Table 2 are the sine wave generator and the final adder. The overall quantity of resources required for sine waves comes from the number of generators (32 independent). This number was chosen to test multi-tone waveform generation and can be reduced to eight generators for simple sinusoidal stimulation in the case of applications such as in [28]. The same reasoning could be applied to the final adder, with an emphasis on the adder complexity which has a quadratic dependence on the number of generators. Its complexity is also related to the very flexible output configuration. System-wide additions are indeed performed at each system clock period, which is oversized in order to fit a pre-defined 20 MHz maximum output frequency. Defining another output sampling frequency would allow subsampling generator output additions and provide only the results that will be sent to converters. This will make adder time-multiplexing possible and further reduce hardware requirements.

Finally, a direct way to optimize the circuit is to design an ASIC instead of a reconfigurable IC, at the expense of development time and cost.

## 5. Conclusions

In this paper, we presented a mixed system architecture for stimulating excitable tissues during in vitro/in vivo experiments. This specific setup is capable of delivering current controlled waveforms up to 1.2 MHz on different kinds of electrodes with a voltage compliance of $-17.9$ V to 18.3 V. Up to our knowledge, no equivalent stimulator offers such a bandwidth which allows the widest range of stimulation paradigms, including high-frequency blocking stimulation. At low frequencies (under 1 kHz) the maximum load impedance is about 1 M$\Omega$. At 1.2 MHz, the maximum load is 1 k$\Omega$. We demonstrated the ability of this system to successfully generate the non-conventional waveforms discussed in the literature. To pave the way to specific and optimized implants, we quantified the system performances in terms of linearity and digital resources consumption. This system will be used in further experiments and will help testing optimal architecture for future design of implantable systems using non-conventional stimulation waveforms.

**Author Contributions:** F.K. and J.C. were involved in ASIC and analog design. J.C. and Y.B. oversaw digital design. G.N. worked on power supply and board design. F.K., Y.B. and L.R. were in charge of tests and validation. S.R. and N.L. worked on conceptualization, project management and secured funding. All authors participated in writing—review and editing. All authors have read and agreed to the published version of the manuscript.

**Funding:** This research was funded by the French Agence Nationale pour la Recherche (ANR, CENAVEX (AN13-NEUIC-0001-01) and BioTIFS (ANR-18-NEUC-0002-02), and by CNRS PRIME project SMARTSTIM. The research was also supported by CMP-CNFM Bordeaux.

**Conflicts of Interest:** The authors declare no conflict of interest.

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
