# Peer review of "IC-Based Neuro-Stimulation Environment for Arbitrary Waveform Generation"

_electronics, doi:10.3390/electronics10151867_

Round 1

Reviewer 1 Report

Remove all the underlining from, you can italicize instead.  

“Energy efficient waveforms:”

“Increased selectivity waveforms:”

“High-frequency blocking stimulation (HFBS):”

“Biomimetic/controllable burst stimulation:”

“………………”

Please make this correction in the whole document.

Page 5

“2.2. Materials”

     Extend the title description to be more expressive.

Page 5

“Our primary objective is to produce….”

      It was an objective in the past that was successfully fulfilled. Substitute “is” with “was”.

Page 8, section title

“2.5 Reconfigurable digital architecture”

At the bottom of the page is the subsection title, but the subsection content begins on the next page. Each section and subsection title must be followed by at least 2-3 lines of sentences.

 The same problem on page 15.

“2.5.1. Overview”

    Extend the subsection title in order to be clearer to the readers what kind of overview you refer.

On page 10,

“Pattern players:”

“FIFO outputs:”

“Sine generator:”

Are underline, italicize instead.  

Author Response

We thank the reviewer for the kind remarks and the time spent. All styles have been modified as recommended and some titles have been expanded to clarify their meaning.

Reviewer 2 Report

1) The authors state that their work is aimed in develop "a stimulator capable of delivering arbitrary waveforms to electrodes, and suitable for non-conventional stimulation strategies. ", but why? the reason for that should be also stated briefly in the abstract.

2)Please make more consistent/organic the abstract without the schematic division in "Objectives: ... ",  "Methods: ..."  and 
"Discussion: .."

3) Please include in Fig. 2 also the year 2020 (estimation) 

4) Please split Fig. 13 in (two) figures so that in each page the plots can ref to the respective caption (now plots on 2 pages and caption on the third ) 

5) Is not clear which is the reason of exploring different waveforms for the stimulation. For a better compensation of the different charge during anodic and cathodic phases of the biphasic stimulation? Or other reason related to the stimulation effectiveness? this should be better stressed and clarified. Is clear the description of how to get the different waveforms but not the reason for that. 

6) In the reference list no paper from the same journal. Please explore the paper from Electronics  (i.e. https://doi.org/10.3390/electronics9081198 , 
https://doi.org/10.3390/electronics9071156, https://doi.org/10.3390/electronics9050812 among the others)
) or MDPI in general that deal with the same topic and include them in the reference list. Notice that no pages and reference limits in MDPI publication, so feel feel to add every suitable reference.

Author Response

We thank you for your time and remarks. We addressed the different points, as suggested in your comments:

  • Points 1 and 5: A sentence has been added in the Abstract (Such a system enables…, page 1) as well as in the Introduction (We intend to use…, page 4) to clearly state our motivation in designing such a stimulator. As explained in the introduction, this system is designed for in-vivo experiments on different pathologic context, where the stimulation performances are optimized (functional efficacy or energy efficiency) by tuning the waveshape.
  • Point 2: the subsection titles have been deleted in the abstract.
  • Point 3: Figure 2 a has been updated with 4 papers from 2020 showing results from stimulation ASIC (0.18 um and 0.35 um processes).
  • Point 4: Figure 13 has been split in 3 figures, so that captions are near the related curves. References in the text have been updated as well.

Point 6: reference 45 is now on Tala and Johnson (Electronics 2020), which paper deals with the same topic. This paper is cited in the introduction section, in the paragraph listing existing systems for arbitrary waveforms generation (page 4).  

Reviewer 3 Report

A mixed system architecture is presented to stimulate excitable tissues during different types of experiments. The study is a significant one in terms of the designed system with a High Voltage CMOS ASIC generating a configurable stimulating current, driven by a digital circuitry implemented on a FPGA. The stimulator is considered to be applicable and useful in testing unconventional stimulation waveforms and be used for the purpose of generating proof-of-concept data before designing implantable and application-dedicated implantable stimulators.

The novel aspect of the paper is the offering of a maximum versatility on the stimulus waveform. The related application steps of the system proposed are presented and depicted well and in good detail. The references are up to date and relevant as well. Future directions presented are also worthy of investigation.

Yours sincerely,

Author Response

We would like to thank the reviewer for his time and work.